# EEG Correlates of Cognitive Functions in a Child with ASD and White Matter Signal Abnormalities: A Case Report with Two-and-a-Half-Year Follow-Up

**DOI:** 10.3390/diagnostics13182878

**Published:** 2023-09-08

**Authors:** Milica Ćirović, Ljiljana Jeličić, Slavica Maksimović, Saška Fatić, Maša Marisavljević, Tatjana Bošković Matić, Miško Subotić

**Affiliations:** 1Cognitive Neuroscience Department, Research and Development Institute “Life Activities Advancement Institute”, 11000 Belgrade, Serbia; m.cirovic@add-for-life.com (M.Ć.); s.maksimovic@add-for-life.com (S.M.); s.fatic@add-for-life.com (S.F.); m.marisavljevic@add-for-life.com (M.M.); m.subotic@add-for-life.com (M.S.); 2Department of Speech, Language and Hearing Sciences, Institute for Experimental Phonetics and Speech Pathology, 11000 Belgrade, Serbia; 3Department of Neurology, Faculty of Medical Sciences, University of Kragujevac, 34000 Kragujevac, Serbia; stmatic769@gmail.com; 4Clinic of Neurology, University Clinical Centre of Kragujevac, 34000 Kragujevac, Serbia

**Keywords:** EEG correlates, cognitive functions, autism spectrum disorder, white matter signal abnormalities

## Abstract

This research aimed to examine the EEG correlates of different stimuli processing instances in a child with ASD and white matter signal abnormalities and to investigate their relationship to the results of behavioral tests. The prospective case study reports two and a half years of follow-up data from a child aged 38 to 66 months. Cognitive, speech–language, sensory, and EEG correlates of auditory–verbal and auditory–visual–verbal information processing were recorded during five test periods, and their mutual interrelation was analyzed. EEG findings revealed no functional theta frequency range redistribution in the frontal regions favoring the left hemisphere during speech processing. The results pointed to a positive linear trend in the relative theta frequency range and a negative linear trend in the relative alpha frequency range when listening to and watching the cartoon. There was a statistically significant correlation between EEG signals and behavioral test results. Based on the obtained results, it may be concluded that EEG signals and their association with the results of behavioral tests should be evaluated with certain restraints considering the characteristics of the stimuli during EEG recording.

## 1. Introduction

This article presents an extended examination of a young individual diagnosed with autism spectrum disorder (ASD) and white matter signal abnormalities (WMSAs) who was enrolled in two and a half years of continuous integrative therapy. The ultimate goal of this research was to estimate the relationship between electroencephalography (EEG) findings during auditory–verbal processing and auditory–visual–verbal processing with behavioral tests in order to determine to what extent EEG signals may be used to predict and understand sensory, cognitive, and speech–language development in a child with a neurodevelopmental disorder and WMSAs.

Autism spectrum disorder (ASD) is a complex neurodevelopmental disorder that includes deficits in social interaction and communication with associated patterns of restricted and repetitive behavior. While analyzing the wide range of manifestations in ASD children, the neurodiversity principle should be considered, which explains how the autism phenotype emerges out of the interplay between biology and environment in early life [1]. Consequently, children with an ASD phenotype may exhibit various symptoms, including social, emotional, communication, and processing difficulties [2]. More specifically, ASD identification is determined by two factors that must be evident in the area of various possible manifestations that may differ depending on gender and chronological age: (a) continuing difficulties in establishing and retaining social communication and reciprocal social relationships that exceed the expected range of typical functioning considering the individual’s age and intellect and (b) continuous restricted, repetitive, and rigid patterns of behavior, interests, or actions whose nature is clearly excessive or abnormal for the individual’s age and sociocultural context, including sensory hypersensitivity and hyposensitivity [3]. Based on such an ASD identification, it is noticeable that ASD symptoms affect personal, family, social, educational, occupational, or other crucial aspects of functioning significantly [3]. Furthermore, autism’s complexities and heterogeneity have led to a wide range of hypotheses about its causes, but it is certain that autism has multiple causes that occur in various combinations [4]. Recent research has revealed that white matter (WM) development may be atypical in children with ASD [2,5,6,7]. Although these and similar studies confirmed WM structural abnormalities in ASD children, there is diversity among the findings, making it hard to interpret and compare the results. Additionally, some authors [8] indicate the presence of potential variations in neural mechanisms between children and adults. However, the results of studies indicating abnormalities in the structure of white matter in children with ASD are highly significant. They enable us to understand this pathology better and can serve as clinical markers in diagnosing this neurodevelopmental disorder.

The white matter lies beneath the gray matter cortex comprising over half the human brain [9]. It is comprises neuronal fibers coated with an electrically insulated substance known as myelin. The myelin sheath, with varying degrees of myelination [10], enables electrical impulses to transmit quickly and efficiently along the nerve cells. It impacts the processing of information by overseeing the speed and coordination of impulse transmission between distant areas of the cortex [11]. Five white matter fiber pathways have been recognized to play a role in different phases of sensory processing: the superior corona radiata, the centrum semiovale, the inferior longitudinal fasciculus, the posterior limb of the internal capsule, and the splenium [12].

ASD with abnormalities in the WM is primarily characterized by abnormalities of the corpus callosum [13,14,15]. Reduced fractional anisotropy (FA) was found in the splenium [15], the genu [16], and the body [17] of the corpus callosum. This finding may provide insight into social and cognitive impairments in children with ASD, considering that the splenium displays associations with IQ (nonverbal), adaptive functions, and cognitive and executive functions [18]. On the other hand, the genu plays a crucial role in facilitating cognitive and social functions by connecting the prefrontal and orbitofrontal regions. According to the theory of mind (ToM), the neural basis of an essential social function is connected by the genu (anterior cingulate, dorsomedial, and ventromedial prefrontal cortex). Consequently, abnormalities in this corpus callosum region could be linked to the social dysfunction typical of ASD [19]. Furthermore, increased mean diffusivity (MD) in the splenium [17] and the central regions of the corpus callosum [16] can be observed in ASD individuals. The presence of reduced FA and increased MD may indicate axonal damage [20], the loss of WM coherence [21], or demyelination. Notably, the reduction in diffusion anisotropy within the corpus callosum could potentially explain deficient organization or insufficient maturation of projection, commissural, and associative fibers, which contribute to alterations in cognitive processes such as calculating, reading, and working memory, all of which rely on the efficient transmission of information between the cerebral hemispheres [22]. Additionally, there is a notable compromission of the dimensions of the corpus callosum; they are significantly reduced in ASD patients [23]. This reduction in size can be associated with altered cognitive, behavioral, and sensory characteristics in individuals with ASD.

Apart from the corpus callosum, other WM structures are also compromised in ASD individuals. Diffusion tensor imaging (DTI) data demonstrated higher FA values and lower MD values in the cingulum [24]. These findings correlate with atypical behaviors related to pain perception, empathy, and social behavior observed in individuals with ASD, as the cingulum is involved in the execution of these functions. Therefore, the altered white matter structure of the cingulum may be associated with the atypical behaviors observed within this diagnosis.

Brain images of ASD children indicate the increased volume of WM in a few brain regions, namely the right insular cortex, right superior temporal gyrus, right Hersch’s gyrus, and left middle temporal gyrus [24], and reduced thickness in the callosal pathways within the middle regions of the brain [2]. Furthermore, ASD individuals have a higher prevalence of short-range u-fibers in the frontal lobe and thinner corticopontine pathways with smaller terminal areas. Corticopontine pathways are known to be involved in saccadic eye movements, a type of rapid eye movement. As individuals with ASD commonly experience impairments in eye movements, the observed alterations in the corticopontine pathway offer a potential explanation for this type of abnormality among ASD patients. In addition, the hyperconnectivity theory of autism suggests that the cause of this neurodevelopmental disorder may be explained by disrupted long-distance connections throughout the brain [25]. Additionally, there is supporting evidence indicating that ASD typically exhibits variations in neural integration that manifest in infancy [26], indicating the importance of preserved brain connections.

In attention-deficit hyperactivity disorder (ADHD) patients, similar to those with ASD, the corpus callosum exhibits the most frequent alteration [13,14]. Notably, the most prominent changes in the corpus callosum, characterized by decreased FA, were observed in the splenium in ADHD and ASD patients [16]. These shared reductions in FA suggest potential common atypicality within a posterior interhemispheric circuit in these neurodevelopmental disorders. Furthermore, ADHD and ASD manifested abnormalities in the corona radiata (CR), distinguishing the two conditions. In ASD, the abnormalities were predominantly found in the anterior parts of the CR, while in ADHD, they were primarily observed in the posterior CR [16]. These findings indicate that WH abnormalities are widespread in both disorders; however, in ADHD, they predominantly affect the posterior brain regions, whereas in ASD, the anterior regions are more affected.

Additionally, certain aspects of executive function impairment are evident in ADHD and ASD [27]. In ASD, compromised social interaction and social knowledge are present, while in ADHD, only social interactions are impaired [28]. This discrepancy could be explained by broader abnormal WH patterns in individuals with ASD, which might impact the manifestation of symptoms.

Like other neurodevelopmental disorders, there are overlapping WH abnormalities in developmental language disorder (DLD) and ASD. Specifically, in both conditions, the volume of WM is increased in the frontal and temporal lobes [29], and in children with DLD, increased WM volume was found in the dorsal striatum [30]. These findings suggest the presence of abnormal myelinization in both disorders. Furthermore, reduced WM volumes within the motor network of the left hemisphere were found in DLD children [31]. These reductions impact the motor cortex, the dorsal and ventral premotor cortex, and part of the superior temporal gyrus called the planum polare [31]. The identified alterations suggest impairments in both motor and language areas within the left brain hemisphere, implying a shared anatomical basis underlying the deficits in these functions.

Generally, it can be concluded that WM plays a crucial role in learning and information processing, but it is also involved and associated with neurological and psychological disorders. Specifically, WM deficiencies have been linked to a variety of psychiatric disorders, including schizophrenia, chronic depression, bipolar disorder, obsessive–compulsive disorder, and posttraumatic stress disorder, as well as neurodevelopmental disorders such as autism spectrum disorder, attention-deficit hyperactivity disorder, developmental language disorder, and dyslexia [11,13,14,15,29,30].

These findings highlight the significance of WH pathways in understanding neurobiological mechanisms underlying ASD-related neurodevelopmental abnormalities. Also, all of the above points to the essential role of white matter maturation in postnatal brain development [32]. Finally, WM has been associated with cognitive differences in humans [33]. 

In recent years, special attention has been given to neuroimaging techniques for identifying locations of brain regions and their specific activation. Most commonly used are functional magnetic resonance imaging (fMRI), computerized tomography (CT), Positron emission tomography (PET) [34,35], and similar techniques with excellent spatial resolution, but they are very expensive and unavailable without experts in this field [36]. Electroencephalography (EEG), due to the development of new methods for EEG signal analysis, has begun to provide new insights into the field of cognitive neuroscience. The advantage of using EEG lies in its good temporal resolution [37]. This method is noninvasive and convenient for measuring the electrical activity of brain regions [38].

EEG findings that may be correlated with ASD are as follows: In a resting-state condition, the frontal and temporal regions decrease in the theta [39] and alpha spectral power [40]. In contrast to this finding, other studies documented increased alpha activity in frontal regions in infants [41], increased theta activity in the left frontal region [42,43], higher spectral power in the right frontal theta [44], and higher gamma spectral power [45]. Likewise, other investigators [46,47] also found an increase in relative frontal theta and relative parietooccipital beta and a decrease in relative alpha in the mentioned regions in a resting-state condition. Also, a more recent coherence study reveals a more intricate connectivity pattern among children with ASD, detecting disruptions in the balance between long- and short-range connectivity within the theta and alpha frequency ranges [48]. 

Moreover, EEG findings related to ASD children and overlapping white matter abnormalities revealed the occurrence of an atypical and slow white matter maturation pattern [49], with decreased alpha over time [50,51], primarily localized in the frontal and posterior regions, and the corpus callosum [49]. 

Because EEG signals and their association with the results of behavioral tests may be sources of valuable information having a profound impact on the diagnosis, treatment planning, and prognosis in children with neurodevelopmental disorders [52], we aimed to broaden our understanding of how much EEG signal may be used to predict sensory, cognitive, and speech–language development in a child with neurodevelopmental disorder and WMSAs. Accordingly, we report the ADOS score; sensory profile; and cognitive, speech–language, sensorimotor, and socio-emotional profiles correlated with EEG findings of a boy exhibiting both ASD and WMSAs, observed in an extended study conducted over the period from 38 to 66 months of age. 

More precisely, the study’s main goal was to correlate the EEG findings of auditory–verbal and auditory–visual–verbal processing with behavioral tests to determine to which extent EEG signals may predict sensory, cognitive, and speech–language development in a child with ASD and WMSAs. This information is valuable to therapists and useful in diagnosis, treatment planning, and prognosis.

## 2. Case Presentation

### 2.1. Case Report

The boy is the first child of healthy, no consanguineous parents (mother 38 years, father 42 years). He lives in a monolingual family where the primary language spoken is Montenegrin. When he was 38 months old, his parents brought him to the Institute for Experimental Phonetics and Speech Pathology (IEPSP) in Belgrade, Serbia, for an examination, and integrative therapy began immediately. A multidisciplinary admission team examined the child: a speech–language pathologist, a psychologist, a psychiatrist, and a neurologist constituted the team. The team first conducted the assessment, followed by individual assessments of each team expert. Anamnestic data collection began with interviewing parents and examining medical records, followed by child observation and administering specific diagnostic procedures. The anamnestic data were collected through elaborate discussions with the parents and by reviewing the child’s medical documents (the Labor and Delivery note, as well as the assessment of general physicians, neurologists, psychiatrists, and otolaryngologists). Due to breech presentation, he was born via cesarean section at 37 weeks of gestation. There were no complications during pregnancy. Apgar score was 9/10, birth weight 3090 g, and birth length 53 cm. Pre- or perinatal risk factors were absent, except discreetly icteric skin (bilirubin below the limit for phototherapy). Prelingual speech phases were reduced, especially the babbling phase, and the lingual phase was significantly delayed. Early motor development was discretely atypical; the boy did not crawl and walked at 18 months. Specific family heredity did not exist.

The audiological findings were contradictory during early childhood. Specifically, when the boy was 29 months old, a hearing evaluation was conducted using the brain stem evoked response audiometry (BERA), which determined a deficit of auditory perception in the right ear and a regular hearing threshold for the left ear. The boy was diagnosed with hyperacusis sensorineuralis unilateralis l dex. He received a hearing aid for his right ear. At 36 and 38 months, the repeated BERA corresponded to an average hearing threshold bilaterally (BERA performed according to the UK protocol by clicking on 40 and 70 dB and tones with a burst of 4 KHz at 40 dB), showing an excellent morphology with regular absolute and inter-wave latencies and all the time synchronous impulse transmission. After repeating the BERA test two times, it was determined that the hearing threshold is within the limits of social contact on both ears and that there is no need for auditory amplification. According to the otolaryngologist, the first finding of the BERA test, which revealed auditory perception deficit in the right ear, was possibly influenced by the instability of the EEG at the time, which is supported by the finding of the magnetic resonance tomography (MRT) of the endocranium revealing visible scar changes more on the left side. The reaction to sound was present upon admission to the IEPSP; he detected the sound but did not localize it. A faster response was to the melody of a children’s song. 

Neurological status at 30 months: cranial nerves without breakouts, neck free. The upper and lower extremities were eutonic, and motility was orderly. Gross motor skills were preserved. Myotatic reflexes were with symmetrical responses. Gait was normal, occasionally on tiptoes. MRT (T2 AX, T1 AX, FLAIR AX, T2 SAG, T2 COR, DWI, SWI, and T1MPR tomograms of the endocranium) was performed, which revealed visible small scarring lesions in the brain parenchyma, supratentorial, bilateral periventricular, and peritrigonal regions, with the largest individual diameter of about 5 mm.

Neurological status at the age of 66 months was as follows: The myotatic reflexes on the upper and lower extremities were regular, the tone was variable, and the tropism was preserved. Sensory integration dysfunction was present. Axial muscle hypotonia was present, with inadequate motor control for the lower extremities and postural reactions with varus of both feet, more to the left. Walking on tiptoes was observed occasionally. The weakness of the orofacial musculature was observed. At 60 months, the MRT findings were similar without significant changes related to the first finding. 

At admission, the child displayed limited verbal expression, resisted cooperation, and exhibited symptoms consistent with ASD. Unreasonable laughter and squealing were present. He looked down at his hands at eye level as he spun in a circle and walked in a straight line. Motor stereotypes were present. Walking on tiptoes and bouncing were observed. The boy examined the toys tactilely and visually and threw them after some time. The functional play was minimally present. Speech comprehension was reduced. His ability to comprehend speech was at the level of a few uncomplicated experiential tasks facilitated through situational cues and gestures. Expressive speech deviated significantly from age norms. According to the parents, the boy produced about five words with uncertain meaning functions. The boy also tended to “forget” a word once used. The syntax was missing. The central voice was variable in intensity and partly in tonality. The demonstrative gesture was absent. Graphomotor ability was not developed. 

At the age of 66 months, the child became verbal. Verbal production was at the level of individual words that occur spontaneously and inconsistently. He mainly produced babbling and idioglossia. There was consistent echolalia in the form of two-syllable and three-syllable words with meaning when he listened to the questions. He demonstrated consistency in concept naming, maintained by constant initiative and encouragement from the therapist. Syntactic development had begun. Sometimes, when encouraged by the therapist who gave him phonological support, he produced short sentences. Verbal imitation was improved. Vowels, plosives (voiced and voiceless), affricate C/ts/, and fricatives S/s/, Z/ʒ/, and V/v/ were present in the phonological capacity. He followed the therapist’s demonstrative gesture more often and more consistently. When not engaged in the task, he frequently used strained, stereotypical, and dysfunctional vocalizations such as chirping or syllabic production. Although the motor stereotypes were still present, the child demonstrated a reduction in stereotypical behaviors and/or unusually recurring interests at 60 and 66 months. The graphomotor ability was minimally improved and was in line with improvements in visuomotor coordination. The drawing was at the level of scribbling. The attention deficit persisted—he did not focus on the task, “fixated” the therapist with his gaze, and/or looked into space.

The multidisciplinary admission team in the IEPSP diagnosed ASD after observing and analyzing the obtained data. Considering all of the information gathered about the child, the cross-disciplinary admission team, in cooperation with a cross-disciplinary implementation team, developed an integrative therapy plan based on the KSAFA approach [53]. The therapy was set to maximally exploit the potential and fulfill the child’s unique needs during treatment. Individual treatment based on KSAFA principles was chosen because it proved effective in children with complex disorders [52,53]. The utilized integrative therapy involved interventions for acquiring communication and linguistic skills, psychomotor reeducation, techniques to enhance sensory processing, and psychotherapeutic support for the child’s parents. 

Despite intensive treatment, the boy demonstrated poor and insufficient improvement in the observed developmental aspects during the two-and-a-half-year integrative therapy. 

### 2.2. Study Design 

At the initial time point (t0), prior to the commencement of the treatment, an initial evaluation was conducted: (1) Sensory Profile 2; (2) Autism Diagnostic Observation Schedule—Second Edition (ADOS-2); (3) cognitive assessment (The Čuturić Developmental Test (RTČ-P) and REVISK); (4) the Scale for Evaluation of Psychophysiological Abilities of Children (SEPAC); and (5) EEG recording in three conditions, namely (a) the resting state, (b) during auditory–verbal stimulation (“listening to a story”), and (c) during auditory–visual–verbal stimulation (watching the animated movie “Peppa Pig cartoon series”).

Treatment started a few days after the initial evaluation. The evaluation was repeated in five test periods (t0 = 38 months; t1 = 44 months; t2 = 54 months; t3 = 60 months; and t4 = 66 months), during continuous integrative therapy. Sensory profile; ADOS score; and cognitive, speech–language, sensorimotor, and socio-emotional profiles were evaluated and compared with EEG correlates during auditory–verbal processing (listening to a story) and auditory–visual–verbal processing (watching and listening to a cartoon). 

The complete study protocol had been approved by the Ethics Committee of the Institute for Experimental Phonetics and Speech Pathology (date: 14 January 2021, No 1/21-4) in Belgrade, Serbia, which operates in accordance with the Ethical Principles in medical research involving human subjects, established by the Declaration of Helsinki 2013. The child’s parents provided written informed consent to participate in this study. 

### 2.3. Data Collection

(1) Sensory Profile 2 [54] is a standardized tool for evaluating children’s sensory processing patterns. Its objective is to pinpoint how a child’s sensory processing influences their day-to-day performance within the home, school, or community settings. It is an 86-item questionnaire for carers and parents. The items are evaluated using a Likert scale (1–5). The sum of the scores on each subscale represents sensory dysfunction. Lower scores indicate less frequent behavior, while higher scores indicate more frequent behavior. The questionnaires consist of diverse scores encompassing sensory systems, behavior, and sensory patterns. Their interpretation aligns with Winnie Dunn’s sensory profile theory [54]. This test was administered at all assessment points.

(2) Autism Diagnostic Observation Schedule—Second Edition (ADOS-2) [55] is used for the assessment of autism-related behavioral patterns in children. ADOS-2 is a gold standard for assessing ASD symptoms in toddlers, children, and adults. It is a semi-structured assessment tool divided into modules based on the participants’ age and degree of language development. Module 1 was used in this study because of the age at which the assessments were performed and due to language production (nonverbal participant). Trained and certified clinicians conducted the assessment following the standardized ADOS-2 protocol. The assessment was performed in a room equipped only with the necessary furniture (three chairs for the child, parent, and clinicians, a table, and a cabinet) to avoid distracting the child’s attention from the testing materials. During the assessment, the child was presented with various social and play-oriented activities, during which the clinicians noted the child’s behavior and reactions. Scoring is based on assessing behaviors exhibited by the child during the evaluation. These behaviors are described and categorized into five domains: language and communication; reciprocal social interaction; play; stereotyped behaviors; and restricted interests and repetitive behaviors. Assigned scores (0, 1, 2, and 3) are subsequently converted into the algorithm: A score of 3 is transformed into a result of 2 in the algorithm; scores 0, 1, and 2 are directly transferred to the algorithm, while codes 7, 8 and 9 are converted into a result of 0 in the algorithm. The algorithm is divided into two categories: social affect (divided into subcategories of communication and reciprocal social interaction) and restricted and repetitive behavior. The overall test score is the sum of these two categories. Based on the obtained score and whether the child is nonverbal/minimally verbal or verbal using multiple words, the child could be classified into nonspectrum, autism spectrum, or autism. This test was also administered at all assessment points.

(3) The cognitive profile was assessed using two different instruments (RTČ-P and REVISK) depending on the child’s age at a specific time point. 

The Čuturić Developmental Test (RTČ-P) [56] evaluates the psychomotor development of infants, toddlers, and preschoolers. It assesses the progression of psychomotor abilities, oculomotor skills, emotional responses, speech, auditory–motor responses, communication and sociability, and verbal knowledge expression. RTČ-P is designed for children aged 2 to 8 and comprises seven subtests, each containing six tasks. There is a consistent thread of tasks involving the manipulation of individual objects across the subtests. The coefficient of mental development expresses the child’s achieved result. The test employs a variety of materials (rattle, bell, pot, bottle, and ball). During the first three assessment points (t0, t1, and t2), the evaluation was carried out in the presence of a parent. 

REVISK [57] is a Serbian revised version of the Wechsler Intelligence Scale for Children. Following Wechsler’s principle, this assessment is standardized to evaluate children’s overall intellectual performance and cognitive skills. The tool offers overall, verbal, and performance scores, where higher scores correspond to higher levels of cognitive functioning. The Verbal and Performance Scales comprise five subtests. Verbal Scale subtests are Information, Comprehension, Arithmetics, Similarities, and Digit Span. Performance Scale subtests are Picture Completion, Picture Arrangement, Block Design, Object Assembly, and Coding. The examination was conducted at the t3 and t4 evaluation points.

(4) The Scale for Evaluation of Psychophysiological Abilities in Children (SEPAC) is used to determine a child’s level of speech and language, sensorimotor, and socio-emotional development relative to the chronological age, measured in months. It includes an assessment of the child’s language development (language production and understanding), sensorimotor development (gross and fine motor skills), and socio-emotional development. The SEPAC is divided into subscales for each year of life. The level of speech and language, sensorimotor, and socio-emotional development is estimated using a subscale specific for chronological age. These metrics are widely used in speech and language clinical work within Serbia [52,58,59,60]. The assessment was conducted at all evaluation points. 

(5) EEG Recordings:

EEG recording was conducted with the child seated comfortably in a soundproof and electromagnetically shielded room. The participant was insulated from visual and auditory stimuli using white curtains arranged in a box-like enclosure within a tranquil environment.

The EEG recordings were acquired using the Nihon Kohden Corporation EEG 1200 K Neurofax apparatus with Electrocap, International, Inc., Ag/AgCl ring electrodes filled with electroconductive gel, providing 19 EEG channels. Electrodes were positioned based on the 10/20 placement system in the longitudinal, monopolar arrangement. The reference electrode was set offline to A1 and A2 (ear lobes). The horizontal and vertical electrooculograms (EOG) were recorded to detect eye blinks and eye movements. The heart rate, hand movement sensors, and electrodes for jaw muscle activity were used for offline artifact removal. The AC filter was activated, and the sampling rate was set to 200 Hz.

Impedance was kept below 5 kΩ, and the difference in electrode impedances was below 1 kΩ. The high-pass filter was set to 0.53 Hz, and the low-pass filter was set to 35 Hz to select the frequency ranges of interest and higher cutoff frequencies that might indicate muscle artifacts. According to the International 10/20 system of electrode positioning, the following cortical regions were analyzed: Fp1-Fp2 (frontopolar), F3-F4 (mid-frontal), F7-F8 (inferior frontal, anterior temporal, and frontotemporal), T3-T4 (mid-temporal), T5-T6 (posterior temporal), Fz (frontal midline central), C3-C4-Cz (central), P3-P4 (parietal), Pz (parietal midline central), and O1-O2 (occipital). 

During spontaneous resting state (task 1), EEG recording was conducted for 2 min, during which the participant was instructed to keep his eyes open. The parent assisted the participant in minimizing movements (such as eye blinks or head and limb movements) to eliminate artifacts from raw EEG traces. In the second task (task 2), a 2-minute EEG recording was performed while the participant listened to the simple short story (“The Red Riding Hood”). The child’s role in this task was passive listening. The third task (task 3) was recording a 3-minute EEG while watching and listening to an animated movie (Peppa Pig cartoon series).

Recordings of task 1, task 2, and task 3 were used to determine the mean spectral power values for theta (4–8 Hz), alpha (8–12 Hz), and beta (13–24 Hz) frequencies in all 19 electrodes.

Before analysis, data segments containing noticeable eye blinking, high amplitude, high-frequency muscle noise, and other irregular artifacts were eliminated using ICA (EEGLAB) [61]. We used fast Fourier transform (FFT) to isolate brain rhythms from the raw EEG trace. 

The initial step in the signal analysis involved the selection of three epochs from the entire recorded signals during task 1 (2 min), task 2 (2 min), and task 3 (3 min), taking 10 s from the beginning, middle, and end of each recording. Before applying FFT, each epoch was multiplied by a suitable windowing function (Hanning window) to prevent boundary leakage. Subsequently, FFT was computed to generate spectrograms and amplitude maps of the chosen epoch. The spectral powers of the three epochs were averaged and filtered to derive spectral powers within theta, alpha, and beta frequency ranges. The average spectral power within each frequency range was further used in subsequent statistical analysis. 

The EEG recording was conducted following the established protocol during all evaluation points.

### 2.4. Statistical Analysis

Summary statistics were presented for the achievements on different behavioral assessment scales (Sensory Profile 2, ADOS-2, The Scale for Evaluation of Psychophysiological Abilities of Children, and the achievements on cognitive assessment scales (RTČ-P and REVISK). Bivariate correlation and its statistical significance between EEG findings and behavioral test results were calculated. Statistical Package for the Social Sciences version 22.0 was used. Linear trend functions were obtained using the Excel program.

### 2.5. Results

#### 2.5.1. Sensory Profile 

The findings presented in Table 1 revealed that, during the initial assessment (time point t0), the child’s sensory processing exhibited significant deviations from the norm. Scores of ±1 or ±2 SD are depicted in the quadrants, sensory, and behavioral sections. Subsequently, a gradual trend toward approaching the average values can be observed at various time points. At the last time evaluation (time point t4), the boy’s sensory processing resembled that of typically developing children in all quadrants. However, he still manifested difficulties in the sensory section (tactile and movement domain) and the behavioral section (conduct and attentional domain).

#### 2.5.2. ASD Symptoms and Cognitive, Speech–Language, Sensorimotor, and Socio-Emotional Profiles

Table 2 presents information on the child’s sensory profile, ASD symptoms, cognitive and speech–language proficiency, sensorimotor skills, and socio-emotional development. During the commencement of the applied integrative therapy (time point t0), the child’s sensory processing showed substantial deviations from the norm. Scores of ±1 or ±2 SD are displayed in the quadrants, as well as sensory and behavioral sections. Subsequently, a progressive trend toward converging with average values can be noted across different time points. By the final assessment (time point t4), the boy’s sensory processing resembled that of typically developing children on all quadrants. However, he still manifested difficulties in the sensory section (tactile and movement domain) and the behavioral section (conduct and attentional domain).

Cognitive assessments indicated borderline intellectual functioning at t0, with a tendency to approach below-average intelligence, more precisely, mild intellectual disability (at t4). The progressive decline in VIQ and PIQ scores should be interpreted as the boy’s persistent difficulties in reaching age norms and responding to increasingly complex cognitive demands. 

A rise in scores across the SEPAC subscales for speech and language, sensorimotor, and socio-emotional development assessment signifies advancement in all evaluated skills. Although the child progressed in these functions, especially concerning socio-emotional development, he was still clearly lagging behind his chronological age. When assessing speech–language development, the difference between achievement (calculated in age months) and calendar age was thirty-nine months, thirty-three months when assessing sensorimotor development, and twenty-four months when assessing socio-emotional development.

At all test points, the ADOS-2 scores were higher than the critical value for the autism spectrum disorder. A slight improvement was observed in the level of communication (t3 and t4) compared with the initial testing and tests at points t1 and t2 on using gestures and directing speech to others. In contrast, no improvement was noted in the level of mutual social interaction compared with the initial testing. Compared with the initial tests (t0 and t1), progress in limited and repetitive behavior was observed at point t2, where there was a reduction in mannerisms with hands and fingers. In contrast, the most significant progress was observed at points t3 and t4, where there was a reduction in stereotypic behaviors and/or unusually recurring interests.

Table 2 displays quantitative achievement indicators at various time points.

#### 2.5.3. EEG Findings

Bearing in mind that the trend in time of the difference between the left and right hemispheres in the theta frequency range can be a predictor of speech and language development [52], we calculated that difference as (Fp1 + Fp3 + F7 − Fp2 − Fp4 − F8)/3 for both tasks. Fp1 through F8 are the mean spectral power in the theta frequency range. The time dependence of differences between the left and right hemispheres in the theta frequency range during watching and listening to a cartoon and listening to a story is presented in Figure 1. 

Before analyzing the trend of the ratios of the individual frequency range in the total EEG signal during the observed period, transformations were performed on the averaged spectral powers of each frequency range.

First, the mean spectral power of each frequency range for each measuring electrode is normalized using the mean spectral power of the resting state of the given frequency range of the observed electrode according to the following equation:$${\mathrm{NFrequency}\;\mathrm{range}}_{\mathrm{task}}^{\mathrm{el}}=\frac{{\mathrm{Frequency}\;\mathrm{range}}_{\mathrm{task}}^{\mathrm{el}}}{{\mathrm{Frequency}\;\mathrm{range}}_{\mathrm{Rest}}^{\mathrm{el}}}$$
where
N Frequency range—normalized theta, alpha, or beta frequency range;Frequency range—theta, alpha, or beta frequency range;task—task 2 (listening to a story) or task 3 (watching and listening to a cartoon); el—one of the 19 EEG measuring points (Fp1 …Cz); Rest—resting state. 

After that, we calculated each rhythm’s mean normalized spectral power for tasks 2 and 3, averaging the normalized spectral power across all electrodes. The values obtained in this way were used to calculate the contribution of each of the three frequency ranges to the total EEG signal according to the following formula:(1)$${\mathrm{RFrequency}\;\mathrm{range}}_{\mathrm{task}}^{\;}=\frac{{\mathrm{MFrequency}\;\mathrm{range}}_{\mathrm{task}}^{\;}}{{\mathrm{Mtheta}}_{\mathrm{task}}^{\;}+{\;\mathrm{Malpha}}_{\mathrm{task}}^{\;}+{\;\mathrm{Mbeta}}_{\mathrm{task}}^{\;}}$$
where
RFrequency range—the relative contribution of the frequency range to the total signal (Rtheta, Ralpha, and Rbeta);MFrequency range—the mean normalized spectral power;task—task 2 (listening to a story) or task 3 (watching and listening to a cartoon).

The time dependences of the relative ratios of theta, alpha, and beta rhythms calculated according to Equation (1) during watching and listening to a cartoon and listening to a story are presented in Figure 2 and Figure 3.

#### 2.5.4. Correlations

Pearson bivariate correlation was calculated between all frequency range variables (RThetaC, RAlphaC, RThetaS, RAlphaS, Theta Cartoon, and Theta Story) and behavioral test results (sensory profile, TIQ, ESLD, ESMD, ESED, and ADOS). Statistically significant correlations between EEG and behavioral tests are presented in Table 3.

## 3. Discussion

This paper presents the outcomes of a study that spanned two and a half years, observing a child diagnosed with autism spectrum disorder and white matter signal abnormalities. To determine the extent to which an EEG signal may be used to predict and understand cognitive and speech–language development in a child with ASD and WMSAs, we collected data on the child’s sensory profile; autism symptoms; cognitive, speech–language, sensorimotor, and socio-emotional profiles; and EEG findings. The data were obtained and analyzed in the five assessment points between 38 and 66 months of the child’s age. 

Throughout the monitoring of the development of the examined developmental functions in the child, progress was observed in all functions but with unequal dynamics when viewed from the perspective of chronological age. Specifically, under the conditions of applied integrative therapy, the child demonstrated significant progress in sensory processing but only minor progress in a reduction in autistic symptoms and in speech–language, sensorimotor, and socio-emotional development. Although there was progress in speech–language, sensorimotor, and socio-emotional development, the discrepancy between the current level of functional development and chronological age did not decrease. As previously observed, children with ASD who receive integrative therapy with the determined methodological principle of treatment [52,53] demonstrate significant progress in the observed function’s development. In this study, despite intensive treatment, there was no significant progress when examining developmental functions in the child with ASD and WMSAs compared with his chronological age. In this regard, the question arose to what extent neurological specificities such as WMSAs in children with ASD may influence the treatment dynamics and progress in their developmental functions during the applied integrative and intensive therapy. Additionally, we sought to determine to what extent the EEG correlates of auditory–verbal and auditory–visual–verbal information related to behavioral findings may predict and explain sensory, cognitive, and speech–language development in this neurodevelopmental disorder overlapped with WMSAs. 

### 3.1. Sensory Profile

The child’s sensory profile changed as the therapy advanced. Specifically, at the onset of treatment, the child’s sensory processing exhibited notable deviations from the norm, gradually moving toward average values in subsequent time points (especially at assessment points t3 and t4). It was observed that the sensory capacities were gradually stabilized at points t3 (60 months of age) to t4 (66 months of age), resulting in similar sensory processing in the boy as in typically developing children at the stage of 66 months. Although at the stage of 66 months, the sensory profile improved, the boy still manifested specific difficulties in the sensory section (tactile and movement domain) and the behavioral section (conduct and attentional domain). Such findings are consistent with studies showing that children receiving adequate treatment significantly reduce atypical sensory characteristics [62]. In general, the progress in sensory processing may be explained by the application of sensory integration therapy [63,64,65] and re-education of psychomotor skills (psychomotor training) [66,67]. Given that the sensory profile of the child with ASD and WMSAs improved during intensive treatment, we may assume that children with ASD and WMSAs can significantly improve sensory processing with appropriate treatment.

### 3.2. ASD Symptoms and Cognitive, Speech–Language, Sensorimotor, and Socio-Emotional Profiles

The presence of autism-related behavioral patterns measured by ADOS-2 was significant throughout the follow-up study, with a slight decrease from 44 months of age and again at 60 and 66 months, respectively. More precisely, at assessment points t3 and t4, this slight improvement was measured in the level of communication and the use of gestures. In contrast, no improvement was noted in the level of mutual social interaction. At 54 months, slight progress in limited and repetitive behavior manifested with reduced mannerisms with hands and fingers. On the other hand, the child demonstrated more significant progress in reduced stereotypic behaviors and/or unusually recurring interests at 60 and 66 months. Bearing in mind that the child has an overlapping diagnosis of ASD and WMSAs, it can be assumed that the aberrantly developed WM significantly caused poor responsiveness to treatment and contributed to a weak reduction in autistic symptoms despite intensive treatment. This may be explained by the fact that white matter has a specific neurobiology, especially an extended period of dynamic development. It could explain why the white matter is exceptionally responsive to treatment, especially if specific abnormalities are detected. This is supported by the findings pointing out that the detection of structural abnormalities in the white matter of children diagnosed with ASD is extremely important in helping us to comprehend this condition further and may be used as clinical indicators in identifying this neurodevelopmental disorder [68]. 

Cognitive assessments indicated borderline intellectual functioning at 38 months, with a tendency of continuously decreasing, which resulted in below-average intelligence, more precisely, the category of mild intellectual disability measured at 66 months. This progressive decline in VIQ and PIQ scores, as well as TIQ scores, reflected the child’s persistent difficulties in reaching age norms, that is, responding to increasingly complex cognitive demands. The findings in the literature regarding the cognitive profile of children diagnosed with ASD reveal divergent perspectives. While some argue that ASD is associated with average or high IQ [69,70], others have demonstrated an association with low IQ [70,71,72]. In our follow-up study, in addition to ASD symptomatology, which may be associated with below-average intelligence, the WM abnormalities may also contribute to the observed decline in TIQ, especially if we consider the findings that implicate the role of myelin in cognitive functioning and information processing [11].

The speech–language, sensorimotor, and socio-emotional profiles slightly progressed from the beginning of the treatment until 66 months of age. Specifically, the discrepancy between the current level of estimated functions and chronological age was observed in all the examined functions. Regarding speech–language development, the boy became verbal but still had reduced comprehension and spoken production (inconsistently used individual words when asked to name objects). Speech–language development lagged behind the chronological age, with 31 months of delay at the child’s age of 38 months (the beginning of the examination) and up to 39 months of lag at the age of 66 months. A slight progress was registered in sensorimotor development during the intensive treatment. Specifically, sensorimotor development lagged behind the chronological age by 26 months when the child was 38 months old and by 33 months at 66 months. Although the progress was most noticeable in socio-emotional skills, the difference between achievement (calculated in age months) and calendar age was twenty-four months. This minor progress in speech–language, sensorimotor, and socio-emotional development indicated a trend of certain stagnation regarding the development of the observed functions. Furthermore, research findings indicate strong relationships between white matter development and cognitive development during infancy [73,74,75,76]. Additionally, research has demonstrated changes in white matter development in response to treatment among adults [77,78,79,80] and children [68]. One review study documented white matter neuroplasticity concerning treatment effects in children and adults [81] and similarly in children with disabilities (also, see the review in [82]). By using cognitive–motor intervention in children with disabilities, there is evidence of FMRI changes in white matter neuroplasticity in the frontal, prefrontal, and posterior regions, i.e., the temporal occipital regions and the cerebellum [83,84]. In addition, Maximo et al. [85] provided evidence of the presence of neuroplasticity during treatment; better language skills and improved brain connectivity were observed during reading skills training in ASD children. Clinical and neuroimaging progress after a 12-month treatment intervention in children with ASD and WMSAs was also documented [86]. Although the literature suggests a close relationship between cognition and WM development, indicating WM neuroplasticity due to treatment, our findings revealed the opposite. Specifically, our research findings indicated a slow child’s progress during the applied integrative and intensive therapy, providing a possible explanation that WMSAs, as neurological specificities in children with ASD, may have a significant impact on treatment dynamics and these children’s progress, which is important to consider in the context of the complexity and heterogeneity of autism’s causes and manifestations [4]. Furthermore, the largest longitudinal diffusion-weighted MRI study of white matter development in ASD individuals across early childhood revealed a slower developmental trajectory of FA measures, leading to decreases in FA with age, which indicates that the previously documented differences between cross-sectional studies of younger and older autism cohorts are a result of changes in dynamic biological functions of white matter development [87]. Moreover, significant differences were observed in white matter development between individuals whose autism severity varied, with increased, stable, or decreased severity levels, suggesting that the underlying structure of connections throughout the brain is fundamentally related to autism physical outcomes [87]. Regarding the severity of ASD symptoms, it was reported that, while 54.4% of children with ASD have a stable severity level from approximately 2.5 to 7 years of age, 28.8% have decreased and 16.8% have significantly increased severity levels over this period [88]. In general, our findings are consistent with the divergent data that show how much children with ASD (with or without WMSAs) may differ in terms of treatment effectiveness and progress and measurement procedure outcomes.

### 3.3. EEG Findings

A previous study showed the regularity pattern regarding the difference between the left and right frontal brain regions in theta rhythm during listening to a story, and more precisely, the dominance of the frontal theta rhythm over the right hemisphere’s theta rhythm at younger ages and the left hemisphere’s theta rhythm at the child’s age of 54, 60, and 66 months [52]. This may be explained to some extent by the fact that the child was diagnosed with ASD but without WMSAs or some other neurological disorder, which was present in the child observed and tested in our case study. Bearing in mind that the child in the study of Maksimović et al. [52] and the child in our study were on intensive integrative therapy for two and a half years, the different findings related to the EEG correlates of auditory–verbal processing, and in terms of the children’s progress on the presented behavior tests, lead us to the conclusion that highlights the importance of WM in information processing and cognitive and speech–language development. 

The EEG correlates of auditory–verbal stimulation as the first type of stimulation (listening to a story) and auditory–visual verbal stimulation (listening and watching a cartoon) as the second type of stimulation revealed different results. Specifically, during the first type of stimulation (listening to a story), there was no trend of spectral power ratio changes in the observed frequency ranges (theta, alpha, and beta). On the one hand, there was a positive linear trend (an increase over time in theta spectral power while listening and watching the cartoon) and a negative linear trend (a decrease over time in alpha spectral power); on the other hand, there was no clear trend of spectral power activity at the level of the entire cortex for the beta frequency range.

Certain publications revealed atypical, altered, and absent auditory speech processing in ASD children; see review by Haesen et al. [89]. While in typically developed children, the speech processing maturation pattern goes from bilateral to left-lateralized brain regions [90,91], in ASD children, the right hemisphere seems to be a precursor for speech processing but with an atypical maturation pattern [89]. The absence of narrative processing revealed in our study could be explained by abnormal auditory processing in children with ASD, which leads to the abnormal maturation of language skills [92], more specifically, atypical semantic and syntax comprehension [93], as well as pragmatic difficulties [94] in these children. Furthermore, one review study reported that whole-brain regions have “more randomly organized” activation in resting state and tasks with over-/under-connectivity of brain waves [95]. This fact provides compelling evidence that supports the nonexistence of verified patterns in ASD brain functioning revealed in our study by analyzing the EEG correlates of auditory–verbal and auditory–visual–verbal processing in a child with ASD and WMSAs. 

On the other hand, a significant increase in theta spectral power while listening and watching the cartoon and, conversely, a decrease in alpha spectral power may be explained by the fact that better motivation engagement was found in ASD children during cartoon watching when compared with looking at photos [96,97]. In addition, our results are also in line with a previous case study [98] in which better activation of the amygdala and fusiform gyros to cartoon characters were detected. Given that the amygdala is involved in the processing of emotions [99], our results suggest better emotional involvement in the cartoon than in story listening.

Our findings related to the increased theta and decreased alpha spectral power during watching and listening to the cartoon may have resulted from the presence of visual attention rather than semantic processing in ASD, which is in line with the study by Ceponiene et al. [100], where ASD children were found to have awareness but not semantic approach to auditory stimuli. Furthermore, there is an assumption that delayed processing is a result of attention disorders in ASD [101]. Smith et al. [102] documented visual, rather than auditory, fascination in ASD children. 

In one study on adults with white matter disease, increased theta was correlated with white matter disease, specifically in the limbic system and temporal lobe [103]. By contrast, alpha decreased from the frontal to the occipital region during observations of whole-brain activity. The same author concluded that decreased alpha explained cognitive impairment in adults with white matter disease, while increased theta acts like a brain compensatory mechanism. Similarly, reports of hyperactivation in the theta wave were found in ASD children who did not have WMSAs during different performance tasks [104,105]. Furthermore, Huberty et al. [106] investigated the alpha and theta spectral power in frontal regions’ EEG during resting in children with ASD aged 3 to 36 months. Different trends were obtained in the spectral power of alpha and theta rhythm (alpha increased, while theta decreased with the age of children), compared with the results of our study (alpha spectral power decreased and theta spectral power increased with the child’s age). A possible explanation is the age difference (children aged 3–36 months vs. a child aged 38–66 months in our study). Another possible explanation may be that the study above studied the general ASD population, whereas the child in our study had ASD and WMSAs. Regarding the alpha spectral power, there is no consensus in the literature concerning abnormal alpha wave profiles in patients with autism spectrum disorder observed during resting state. This may be due to patient phenotype variability and the small sample sizes used in the studies [107]. A study examining alpha waves at rest as a neuromarker of autism spectrum disorders found no evidence for abnormal alpha wave profiles in ASD [107]. In contrast, some studies did, particularly those pointing to alpha–mu abnormalities in ASD [108]. On the other hand, in the study that analyzed peak alpha frequency as a neural marker of cognitive function in children with ASD in comparison to typically developing children during resting state, the peak alpha frequency was reduced in ASD children, compared with typically developing children [109]. Moreover, Jan et al. [110] provided evidence of significant brain functional abnormalities in ASD children during exposure to animated movies, such as decreased activation in the frontal and cingulate regions and increased activation in parietotemporal and cerebral regions.

### 3.4. Correlations

When we summarized and analyzed the obtained EEG findings and behavioral tests, statistically significant high and very high correlations were observed between behavioral test results and the relative contribution of the theta and alpha frequency range to the total signal while watching and listening to a cartoon. Specifically, an increase in the proportion of the theta frequency range in the EEG signal correlated with a reduction in sensory profile deviations. In other words, the greater the level of theta frequency range, the better the child’s sensory profile, with lower deviation values. TIQ and ADOS scores negatively correlated with the theta frequency range, while a positive correlation was noted between the theta frequency range and the estimated speech–language, sensorimotor, and socio-emotional development. In addition, there was a high positive correlation between the alpha frequency range and the ADOS score. No correlation was observed between the theta frequency range during listening to a story and estimated sensory profile, speech–language, sensorimotor and socio-emotional development. Such findings may indicate that the observed correlation between the theta frequency range during watching and listening to a cartoon is not directly related to the results of behavioral tests but is related to the increase in the theta frequency range due to visual stimulation and fascination while watching a cartoon, which was also observed in other studies [100,102]. 

In general, in ASD children, there is evidence of atypical sensory processing [111], i.e., less or no activation of the brain structures related to audio–visual processing [112], precisely, parietooccipital [113] and temporal [114] regions. Moreover, atypical theta activation occurs, which may also be related to abnormal brain activation [115]. In particular, such findings, especially if viewed from the perspective of the presence of WMSAs, may explain the results of our study.

This study had some limitations. Firstly, we decided on a case report due to the objective difficulties during the EEG recording of young children with ASD. In particular, younger children with severe ASD are significantly less cooperative and adaptable during EEG recording, directly affecting the increase in EEG trace artifacts. Consequently, artifact removal may significantly affect the level of useful signals used for EEG processing. Also, we initially chose an EEG cap with 19 channels when setting up for EEG recording. Since this is a longitudinal study, the methodological procedure had to remain unchanged. In the later phases of the acquisition process, we did not use a high-density EEG cap with additional channels (32, 64, or 128 electrodes). Also, there is a lack of comparison of EEG findings with the results of other scientists concerning children with ASD and WMSAs. To our knowledge, no longitudinal study dealt with EEG correlates during specific cognitive tasks in children with ASD and WMSAs. As a result, given that this is a case study, future research should include a large research sample of children with ASD and WMSAs to investigate the EEG correlates of cognitive functions in these children. This will enable different approaches in the analysis of EEG signals. It would also be advantageous for the EEG analysis to use a higher density cap with additional channels, at least 32 electrodes, which could highlight the activity recorded in specific regions. Finally, the study’s aim, among other objectives, was to point out the direction toward which future research should be aimed. There is a need for systematic monitoring of children with ASD and WMSAs at the earliest age as it may positively impact treatment dynamics and ASD children’s progress.

## 4. Conclusions

EEG findings revealed no functional theta frequency range redistribution in the frontal regions, favoring the left hemisphere during speech processing. The results pointed to a positive linear trend (an increase over time in the relative theta frequency range during listening and watching the cartoon), and a negative linear trend (a decrease over time in the relative alpha frequency range), while there was no clear trend of the relative beta frequency range at the level of the entire cortex. Conversely, there was no trend of relative frequency range ratio change in the theta, alpha, and beta while listening to a story. Behavioral tests indicated a slow child’s progress during two and a half years of applied integrative and intensive therapy, which may be partially explained by WMSAs. Based on the obtained results, it may be concluded that EEG signals and their association with the results of behavioral tests should be evaluated with certain limitations considering the characteristics of the stimuli during EEG recording. Finally, WMSAs, as neurological specificity in children with ASD, may significantly impact treatment dynamics and the progress in these children, indicating the importance of WM in cognitive and speech–language development and information processing in general.

## Figures and Tables

**Figure 1 diagnostics-13-02878-f001:**
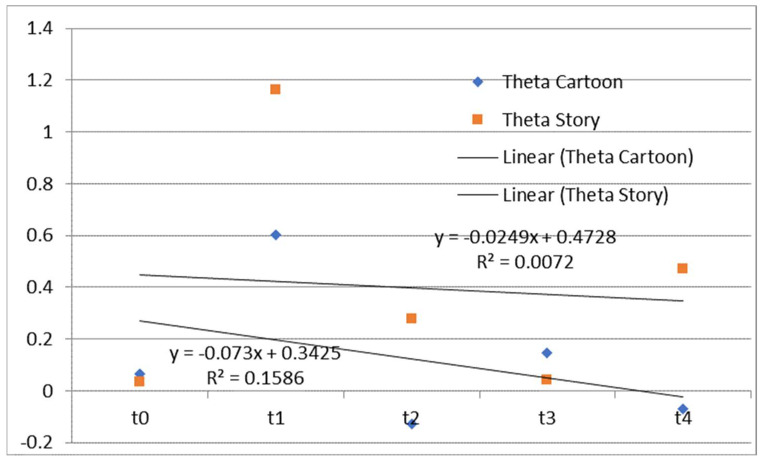
The time dependence of differences between the mean spectral power of the left and right hemispheres in the theta frequency range during watching and listening to a cartoon and listening to a story. Theta Cartoon—differences between the mean spectral power of the left and right hemispheres in the theta frequency range during watching and listening to a cartoon; Theta Story—differences between the mean spectral power of the left and right hemispheres in the theta frequency range during listening to a story; Linear (Theta Cartoon)—linear approximations of Theta Cartoon; Linear (Theta Story)—linear approximations of Theta Story.

**Figure 2 diagnostics-13-02878-f002:**
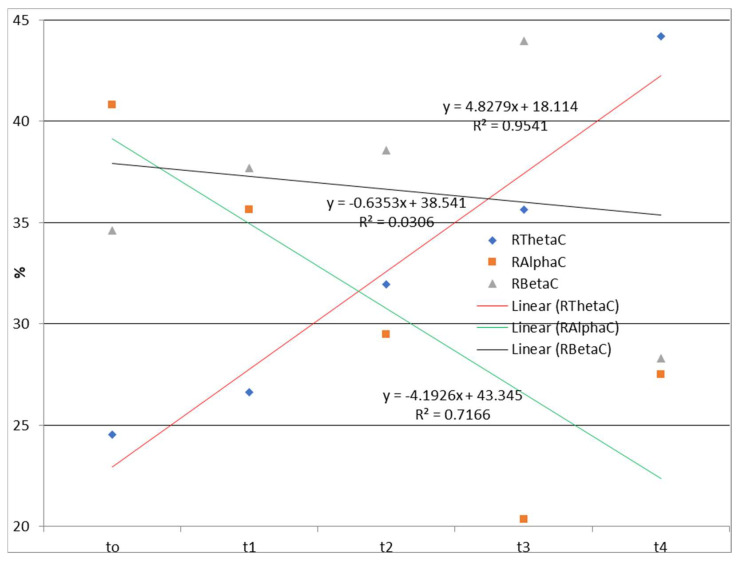
Time dependences of the relative ratios of theta, alpha, and beta rhythms calculated according to Equation (1) during watching and listening to a cartoon. RThetaC—the relative contribution of the theta frequency range to the total signal during watching and listening to a cartoon; RAlphaC—the relative contribution of the alpha frequency range to the total signal during watching and listening to a cartoon; RBetaC—the relative contribution of the beta frequency range to the total signal during watching and listening to a cartoon; Linear (RThetaC)—linear approximations of RThetaC; Linear (RAlphaC)—linear approximations of RAlphaC; Linear (RBetaC)—linear approximations of RBetaC.

**Figure 3 diagnostics-13-02878-f003:**
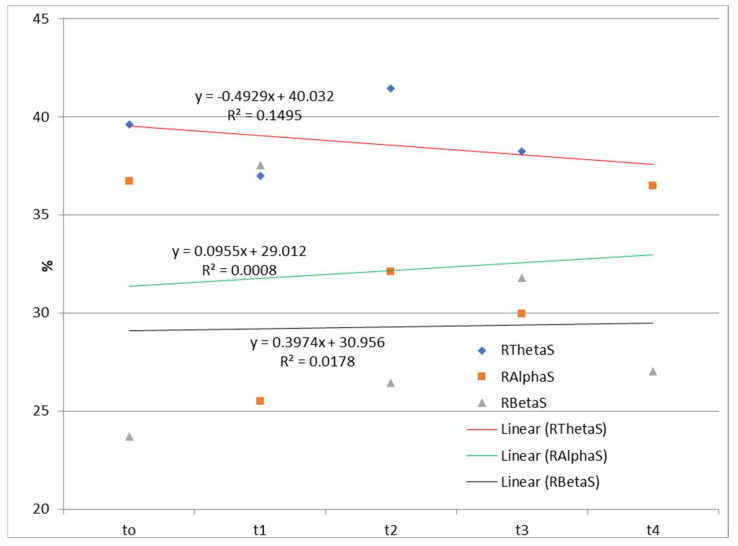
Time dependences of the relative ratios of theta, alpha, and beta rhythms calculated according to Equation (1) during listening to a story. RThetaS—the relative contribution of the theta frequency range to the total signal during listening to a story; RAlphaS—the relative contribution of the alpha frequency range to the total signal during listening to a story; RBetaS—the relative contribution of the beta frequency range to the total signal during listening to a story; Linear (RThetaS)—linear approximations of RThetaS; Linear (RAlphaS)—linear approximations of RAlphaS; Linear (RBetaS)—linear approximations of RBetaS.

**Table 1 diagnostics-13-02878-t001:** Results of Sensory Profile 2 at all the assessment points.

Sensory Profile		Time and Age (in Months)									
		t0		t1		t2		t3		t4	
		38		44		54		60		66	
		Score	SD	Score	SD	Score	SD	Score	SD	Score	SD
Quadrants	Seeking	53	+1	50	+1	39	$\bar{x}$	36	$\bar{x}$	32	$\bar{x}$
	Avoiding	40	$\bar{x}$	40	$\bar{x}$	38	$\bar{x}$	31	$\bar{x}$	24	$\bar{x}$
	Sensitivity	68	+2	65	+2	61	+2	47	+1	35	$\bar{x}$
	Registration	66	+2	65	+2	57	+2	50	+1	42	$\bar{x}$
Sensory section	Auditory	15	$\bar{x}$	14	$\bar{x}$	14	$\bar{x}$	13	$\bar{x}$	11	$\bar{x}$
	Visual	5	−1	5	−1	5	−1	7	−1	9	$\bar{x}$
	Tactile	42	+2	42	+2	38	+2	34	+2	23	+1
	Body position	23	+2	23	+2	21	+2	17	+1	12	$\bar{x}$
	Movement	38	+2	38	+2	24	+2	18	+1	17	+1
	Oral	38	+2	32	+1	23	$\bar{x}$	16	$\bar{x}$	10	$\bar{x}$
Behavioral section	Conduct	15	$\bar{x}$	15	$\bar{x}$	13	$\bar{x}$	11	$\bar{x}$	8	−1
	Social-Emotional	30	$\bar{x}$	30	$\bar{x}$	30	$\bar{x}$	24	$\bar{x}$	20	$\bar{x}$
	Attentional	40	+2	40	+2	40	+2	32	+2	27	+1

Notes: SD—standard deviation from the mean population achievement; +1SD means more than others; −1SD means less than others; +2SD means much more than others; −2SD means much less than others; an $\bar{x}$ average score indicates that sensory processing is just like the majority of other typically developing children.

**Table 2 diagnostics-13-02878-t002:** Results of the sensory profile, cognitive, speech–language, sensorimotor, socio-emotional, and ADOS-2 assessments at different time points.

Assessment Points	Data				
	t0	t1	t2	t3	t4
Age (in Months)	38	44	54	60	66
Sensory profile 2 (Number of standard deviations)	16	15	13	9	4
Cognitive assessment (TIQ) Age in months	70	68	66	63	59
SEPAC- Estimated speech–language development (ESLD) Age in months	7	14	20	20	27
SEPAC- estimated sensorimotor development (ESMD) Age in months	12	23	26	27	33
SEPAC- estimated socio-emotional development (ESED) Age in months	10	11	36	36	42
ADOS score	32	32	31	29	29

Notes: TIQ—total IQ; ESLD—estimated age based on speech and language development; ESMD—estimated age based on the level of sensorimotor development; ESED—estimated age based on socio-emotional development.

**Table 3 diagnostics-13-02878-t003:** Pearson correlation coefficients and obtained statistical significance between the relative contribution of the theta and alpha frequency range to the total signal and behavioral test results.

EEG Theta/Alpha Frequency Range		Sensory Profile	TIQ	ESLD	ESMD	ESED	ADOS
RThetaC	Pearson Correlation	−0.990	−0.994	0.944	0.889	0.896	−0.913
	Sig. (2-tailed)	0.001	0.001	0.016	0.044	0.040	0.030
RAlphaC	Pearson Correlation						0.890
	Sig. (2-tailed)						0.043

Notes: TIQ—total IQ; ESLD—estimated age based on the level of speech and language development; ESMD—estimated age based on the level of sensorimotor development; ESED—estimated age based on the level of socio-emotional development; RthetaC—the relative contribution of the theta frequency range to the total signal during watching and listening to a cartoon; RalphaC—the relative contribution of the alpha frequency range to the total signal during watching and listening to a cartoon.

## Data Availability

Not applicable.

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
