# Peer review of "EEG Correlates of Cognitive Functions in a Child with ASD and White Matter Signal Abnormalities: A Case Report with Two-and-a-Half-Year Follow-Up"

_diagnostics, 2023, doi:10.3390/diagnostics13182878_

Round 1
Reviewer 1 Report
1. The study provides valuable insights into the EEG correlates of stimulus processing in a child with ASD and white matter signal abnormalities during a prolonged follow-up.
2. The research strategy employed in this study was thorough and encompassing, since it involved the collection of data pertaining to a wide range of developmental functions. These functions included the sensory profile, autistic symptoms, cognitive capacities, speech-language skills, sensorimotor abilities, socio-emotional profiles, and EEG findings. Furthermore, data collection occurred at many assessment points during the study.
3. The examination of enhancements in sensory processing over a period of time and the beneficial effects of integrative therapy on the sensory profile of children diagnosed with ASD and WMSA is particularly remarkable in academic literature.
4. This work enhances our comprehension of the significance of white matter anomalies in children diagnosed with ASD and their potential impact on treatment dynamics and developmental advancements.
5. The EEG results during auditory-verbal and auditory-visual-verbal processing give important information about how the brain works and could have effects on how children with ASD and WMSA grow their senses and minds.
1. The research conducted in this study was constrained by a small sample size. As a result, the capacity to apply the findings to a larger group of children with ASD and white matter signal anomalies may be limited.
2. The lack of a control group hinders the ability to compare the progress and EEG correlates observed in the child with ASD and WMSA against typical development or children without WMSA.
3. The study did not investigate potential confounding factors or other comorbidities, such as other neurological conditions or external factors that may have influenced the child's developmental progress and EEG patterns.
4. The inconsistency of the results regarding EEG correlates of auditory processing and cognitive development with prior research findings raises questions regarding the specificity and reproducibility of the observed trends.
5. The paper would benefit from a deeper examination of the specific implications of white matter abnormalities on cognitive functioning and the potential neuroplasticity of white matter in response to treatment.
The quality of English can be improved.
Author Response
Dear reviewer,
Please find enclosed the revised version of the manuscript.
Thank you very much for your valuable comments.
Kind regards,
Ljiljana Jeličić

Reviewer 2 Report
Please emphasize novelty and contribution in Abstract and Instroduction.
Introduction, lines 44-47
ASD is umbrella term, it would be usefeful explainnit based on e.g.
Bach B, Vestergaard M. Differential Diagnosis of ICD-11 Personality Disorder and Autism Spectrum Disorder in Adolescents. Children (Basel). 2023 Jun 1;10(6):992. doi: 10.3390/children10060992.
Whitehouse A. Commentary: A spectrum for all? A response to Green et al. (2023), neurodiversity, autism and health care. Child Adolesc Ment Health. 2023 Sep;28(3):443-445. doi: 10.1111/camh.12666.
Material and Methods, lines 388-409
ML-based methods would be also useful here, see e.g.
Wojcik GM, Masiak J, Kawiak A, Kwasniewicz L, Schneider P, Postepski F, Gajos-Balinska A. Analysis of Decision-Making Process Using Methods of Quantitative Electroencephalography and Machine Learning Tools. Front Neuroinform. 2019 Nov 27;13:73. doi: 10.3389/fninf.2019.00073.
Subsection 3.3. EEG findings: Please provide higher resolution/clarity of Figures 1-3.
Discussion: there is lack of comparison with results of other scientists, please add limitations of the own study and directions for further studies (analysis too).
Author Response
Dear Reviewer,
Please find the enclosed revised version of the manuscript.
Thank you very much for your valuable suggestions.
Kind regards,
Authors

Round 2
Reviewer 1 Report
None